# Self-Expandable Metal Stents for Left Sided Colon Obstruction from Diverticulitis. A Single Center Retrospective Series

**DOI:** 10.3390/medicina57030299

**Published:** 2021-03-23

**Authors:** Antonietta Lamazza, Maria Vittoria Carati, Anna Guzzo, Anna Maria Pronio, Virgilio Nicolanti, Angelo Antoniozzi, Antonio V. Sterpetti, Enrico Fiori

**Affiliations:** Department of Surgery “Pietro Valdoni”, Sapienza, University of Rome, 00161 Rome, Italy; antonietta.lamazza@uniroma1.it (A.L.); mariavittoria.carati@uniroma1.it (M.V.C.); a.guzzo@policlinicoumberto1.it (A.G.); annamaria.pronio@uniroma1.it (A.M.P.); virgilio.nicolanti@uniroma1.it (V.N.); a.antoniozzi@policlinicoumberto1.it (A.A.); antonio.sterpetti@uniroma1.it (A.V.S.)

**Keywords:** self-expandable stents, colorectal obstruction, diverticulitis

## Abstract

*Background and Objectives*: The incidence of diverticulitis is increasing in western countries. Complicated diverticulitis is defined as diverticulitis associated with localized or generalized perforation, localized or distant abscess, fistula, stricture or obstruction. Colonic symptomatic strictures are often treated with segmental colectomy. The aim of our study is to report our experience with Self Expandable Metal Stents (SEMS) placement to relieve sigmoid obstruction secondary to diverticulitis, either as a permanent solution or as a bridge to elective colectomy. *Material and Methods*: From January 2016 to December 2018, 21 patients underwent SEMS placement for sigmoid obstruction secondary to diverticulitis at our institution. In four patients with poor general conditions, SEMS was considered the definitive form of treatment. In 17 patients, the stent was placed as bridge to elective colectomy. Data were prospectively collected and retrospectively analyzed. Primary outcomes were postoperative mortality and morbidity after SEMS and subsequent elective colectomy. *Results*: There was no mortality or major morbidity after SEMS placement or subsequent elective colectomy. No stoma was performed. *Conclusions*: Placement of Colorectal Self Expandable Stent represents a useful tool to relieve obstruction in patients with left-sided colonic diverticulitis. SEMS placement makes it possible to transform an emergency clinical condition into an elective condition, giving time to resolve the inflammation and the infection inevitably associated with complicated diverticulitis.

## 1. Introduction

Diverticular disease of the colon is a common condition; 50% of people over the age of 60 have colonic diverticula [1,2]. Of those with diverticulosis, the lifetime risk of developing diverticulitis is estimated at 10–25%, although more recent studies estimated a 5% rate of progression to diverticulitis [3,4,5]. Eighty-five percent of episodes of acute diverticulitis are uncomplicated (defined by absence of abscess, bowel obstruction, perforation, or fistula formation). The incidence of diverticulitis is increasing in Western countries. Besides increasing age, other risk factors for diverticular disease include the use of NSAIDS, aspirin, steroids, opioids, smoking and sedentary lifestyle [6,7,8]. Diverticulitis can be further sub-classified into complicated and uncomplicated presentations. Uncomplicated diverticulitis is characterized by inflammation limited to the colonic wall and surrounding tissue. Complicated diverticulitis is defined as diverticulitis associated with localized or generalized perforation, localized or distant abscess, fistula, stricture or obstruction. Colonic symptomatic strictures are often treated with segmental colectomy.

Resection of diseased bowel to healthy proximal colon and rectal margins remains a fundamental principle of treatment. A differential diagnosis between obstruction secondary to diverticulitis or secondary to cancer is not always feasible, despite endoscopic and multiple biopsies. Even if surgical resection often appears to be a simple and straightforward solution of the problem, either from a diagnostic or a symptomatic point of view, often, the inflammation of the colon is more extended than expected and management of the patient with an end colostomy (Hartman procedure) may represent a wiser choice. Between 20–50% of patients treated with sigmoid resection and an end colostomy for severe diverticulitis will never be reversed to their normal anatomy. The reasons for high rates of permanent colostomies are multifactorial, including advanced age or the poor general condition of the patient [9,10]. In this situation, the placement of self-expandable metal stents may represent a valid temporary choice, reducing the risks related to the complexity of a therapeutic approach for an acute or subacute clinical condition [11,12]. The stent can be placed in elective conditions, in a proper environment. The stent may represent a definitive solution in high risk patients who present a significantly reduced life-expectancy, or a bridge to an elective resection. After stenting, a complete colonoscopy with multiple biopsies may be performed to exclude the possibility of cancer. The aim of our study was to report our experience with SEMS placement to relieve sigmoid obstruction secondary to diverticulitis, either as a permanent solution or as a bridge to elective colectomy.

## 2. Materials and Methods

### 2.1. Clinical Series

From January 2016 to December 2018, 21 patients underwent SEMS placement for sigmoid obstruction secondary to diverticulitis at our Institution. Data were prospectively collected and retrospectively analyzed. The study was conducted according to the guidelines of the Declaration of Helsinki and approved by the Institutional Review Board of Department of Surgery “Pietro Valdoni”, Sapienza University of Rome (protocol code 154/B, 5 November 2015). Informed consent was obtained from all patients involved in the study. Indication for SEMS placement was decided after a careful analysis of all of the characteristics of the patients by a team composed of gastroenterologists and surgeons. Table 1 shows the clinical characteristics of the patients. Table 2 reports the pathology of the colonic obstruction in 21 patients.

### 2.2. Therapeutic Approach

All patients were admitted with local and systemic signs of inflammation. All patients complained of pain in the left lower quadrant and symptoms referable to subacute or acute colonic obstruction. Ultrasound and CT scans demonstrated bowel dilatation. There was no evidence of a large pericolic abscess. The wall of the sigmoid colon appeared thickened. Intravenous large spectrum antibiotics were administered. All patients were fully evaluated, and any fluid or electrolyte imbalance was corrected before the procedure. A complete cardiologic and respiratory evaluation was performed, and the proper therapy was started.

### 2.3. SEMS Placement

Once the patients were fully re-equilibrated and signs of local and systemic inflammation regressed, partially or completely (48–72 h from admission), a SEMS was placed. The procedures were performed in elective or semi-elective conditions. A low-pressure water enema was performed few hours before the procedure. The procedure was performed with the patient under light sedation with a benzodiazepine at a dosage depending on the patient’s body weight. All procedures were performed by expert endoscopists with an experience of placement of more than 100 colorectal stents. A pediatric nasogastroscope (Olympus GIF N180, Tokyo, Japan) with a diameter of 4.9 mm was used to pass the obstruction [13,14]. With this modification, direct vision of the anatomy and pathology is possible, and the guidewire can be passed above the obstruction through the channel of the nasogastroscope. This has made the procedure much simpler and faster, theoretically reducing the risk of perforation or bleeding. The SEMS apparatus (Precision Stent System Microvasive; Boston Scientific Corp., Boston, MA, USA) was placed at the level of the obstruction through the guidewire previously inserted and finally deployed under fluoroscopic guidance. The length of the stent ranged from 9 to 12 cm. Different stents were used (Figure 1 and Figure 2). Most patients had a covered stent. We prefer in this condition a covered stent because it easier to remove, even if it dislodges more frequently. However, when employed as a temporary solution, the possibility of dislodgment is low.

### 2.4. Follow-Up Evaluation

The patients were followed up by the operators every three months, with a new colonoscopy with multiple further biopsies every six months.

## 3. Results

### 3.1. Early Results (within 7 Days from SEMS Placement)

No mortalities occurred. The technical success rate was 100%, and the clinical success rate was 100%. No perforation or stent dislodgment occurred in the early postprocedure period. One patient had minor bleeding which resolved spontaneously.

### 3.2. Early Results (from 8 Days to 3 Weeks from SEMS Placement)

No mortalities occurred. One patient had bleeding which resolved spontaneously. In one patient, the covered stent dislodged one month after placement. General and local symptoms resolved completely and it was decided not to perform a colon resection and to follow the patient, recommending instead a healthy life-style, including losing weight, stopping smoking and alcohol abuse, and adhering to an appropriate diet: six months from the procedure, the patient is in good general conditions without symptoms.

### 3.3. Six Months Results

#### 3.3.1. SEMS as Definitive Treatment

Four patients had SEMS placement as definitive treatment, avoiding colon resection (4/21 = 19%). Three patients were in poor general conditions with a short life expectancy. After SEMS placement, symptoms resolved completely and it was decided to avoid a colon resection. Two patients died three months after the procedure (one had metastatic lung cancer, and the other had Alzheimer Disease). The remaining two patients are in good general conditions without symptoms at six months after the procedure. The stent migrated externally at one and four months from placement, with complete resolution of symptoms.

#### 3.3.2. SEMS as a Bridge to Surgery

Seventeen patients had subsequent colon resection. The time interval between SEMS placement and surgery was one month in all but one patients (three-month interval). This interval allowed a complete resolution of local and systemic signs of inflammation. Inpatient management consisted on intravenous antibiotics, intravenous fluids, and pain management. Antibiotics covered gram-negative rods and anaerobes and were given for three to five days before switching to oral antibiotics for a 10 to 14-day course. SEMS was placed as soon as initial signs of inflammation regressed. In general, patients were discharged 5–7 days after SEMS placement and re-admitted three weeks later for elective surgery. Typically, defervescence and improvement in leukocytosis was observed after two to four days of hospitalization. Before surgery, a new complete colonoscopy was performed. All patients underwent bowel preparation the day before colonoscopy with polyethylene glycol-electrolyte solution (PEG-ES). A few hours before the procedure, a low-pressure water enema was performed. Surgery consisted of segmental resection with primary anastomosis (laparoscopic procedures were preferred when feasible). None of the 17 patients had a stoma. The postoperative course was uneventful. All 17 patients are in good general conditions with regular bowel movements at six months follow-up. At histology of the resected colon, there was no evidence of tumor.

Overall, there was no mortality and no major complication (ASGE adverse events classification).

## 4. Discussion

Epidemiological studies have shown a 26% increase in hospitalizations for acute diverticulitis and a 38% increase in elective operations from 1998 through 2005 in USA. In Western nations, left-sided diverticulosis is more common, whereas individuals of Asian descent are likely to have the right-sided disease [1,2,3,4].

Diverticulitis is the result of perforations of the diverticular wall. This causes focal inflammation and necrosis of the region, causing perforation. This may promote local abscess formation, fistulization of adjacent organs, or intestinal obstruction.

On physical examination, tenderness to palpation over the area of inflammation is almost always present due to irritation of the peritoneum. A mass may be felt in approximately 20% of patients if an abscess is present. Bowel sounds are usually hypoactive but can be normoactive. Patients can present with peritoneal signs (rigidity, guarding, rebound tenderness) with bowel wall perforation [5,6,7].

Laboratory tests may show leukocytosis and elevation of C-reactive protein. The radiological test of choice for acute diverticulitis is CT of the abdomen and pelvis. Typical findings of acute diverticulitis in CT scans include bowel wall thickening, pericolic fat stranding, pericolic fluid, and small abscesses confined to the colonic wall, as well as contrast extravasation, indicating intramural sinus and fistula formation [15,16,17].

Abdominal ultrasound can accurately diagnose acute diverticulitis, with comparative sensitivity (84% to 94%) and specificity (80% to 93%) to CT. However, ultrasound results are highly operator dependent, and use is limited despite encouraging data, lower cost, and wide availability.

Endoscopy should be avoided in suspected acute diverticulitis due to an increased risk of perforation. Partial bowel obstruction or pseudo-obstruction due to colonic ileus can occur. A patient presenting with symptoms of acute or subacute obstruction from probable diverticulitis should initially be treated conservatively [18,19,20]. The obstruction can depend on the local inflammation with a secondary colonic ileus. Inpatient management of diverticulitis requires intravenous antibiotics, intravenous fluids, and pain management. Again, antibiotics should cover gram-negative rods and anaerobes and be given for three to five days before switching to oral antibiotics for a 10 to 14-day course. Bowel rest is preferred in patients requiring inpatient admission.

Having multiple episodes does not appear to directly increase the risk of complications. It may increase the risk of fibrosis, leading to stricture formation and subsequent obstruction. Approximately 20% of patients with more than two episodes of diverticulitis will experience chronic abdominal pain due to either irritable bowel syndrome or chronic low-grade diverticulitis and recurrent symptoms of intestinal obstruction [21,22,23,24,25,26,27]. These patients may be referred for elective colectomy for symptom control. Elective operations for diverticulitis have increased by approximately 30% since 1998. Indications for elective surgery may also include the fact that the possibility of carcinoma cannot be excluded despite multiple biopsies and sequential colonoscopies.

The mortality rate in uncomplicated diverticulitis is negligible with appropriate conservative therapy. Complicated diverticulitis requiring surgery may lead to death in approximately 5% of patients. Perforation of the bowel with resulting peritonitis increases the risk of death to 20%. Elective surgery has a very low complication rate and a stoma is rarely needed. The patient can return to a normal life-style in a short time In contrast, emergency colectomy for patients with acute symptoms from complicated diverticulitis has a higher complication rate, with a high probability for a stoma, which will be closed in the future in less than 50% of the patients. Closure of end-colostomy stoma, as performed in the so-called Hartman procedure (colectomy and proximal end-colostomy), is difficult for the adherences formed after the first operation.

Conceptually, SEMS placement in patients with inflammatory colon disease like diverticulitis has been considered complication prone, i.e., the stent can dislodge or lead to perforation. In our study, we have shown that SEMS placement, using specific precautions, can be placed to dilate sigmoid obstruction secondary to diverticulitis. Two major aspects should be considered: (1) Placing the stent when the initial inflammation has been partially or completely resolved with appropriate antibiotic and anti-inflammatory therapy; and (2) To have the stent in place only for a relative short period, i.e., one month.

## 5. Conclusions

Placement of Colorectal Self Expandable Stent represents a useful tool to relieve obstruction in patients with left-sided colonic diverticulitis. SEMS placement makes it possible to transform an emergency clinical condition into an elective condition, providing time to resolve the inflammation and infection inevitably associated with complicated diverticulitis.

## Figures and Tables

**Figure 1 medicina-57-00299-f001:**
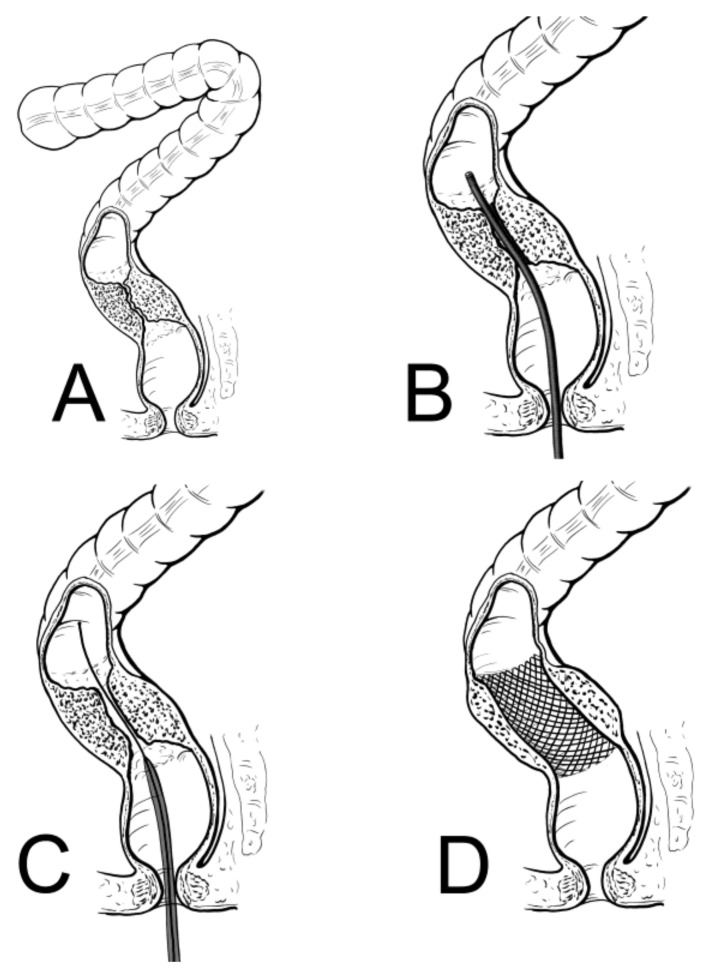
SEMS placement is showed. (**A**,**B**) A thin nasogastroscope is used to pass the obstruction under direct vision to avoid complications (perforation). (**C**) Through the nasogastroscope a guidewire and the stent are placed through the obstruction. (**D**) Appropriate placement of the stent.

**Figure 2 medicina-57-00299-f002:**
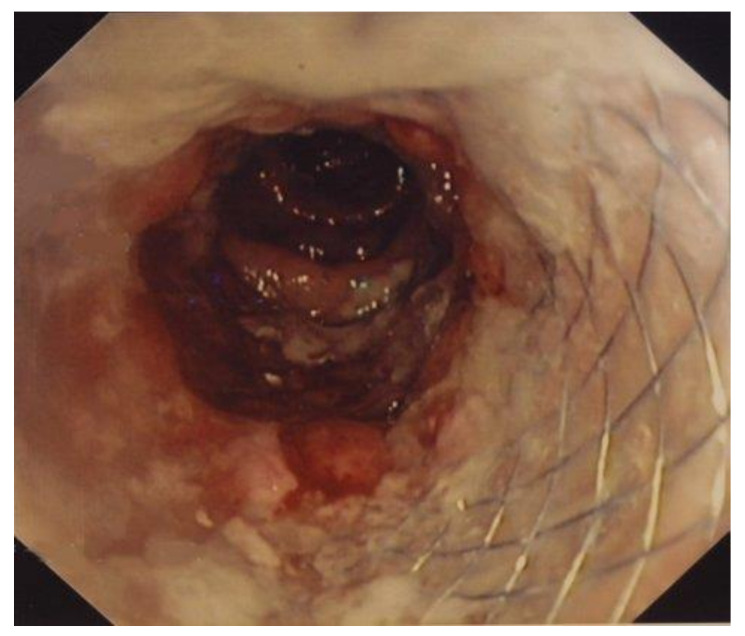
Colonoscopy showing the correct placement of the SEMS.

**Table 1 medicina-57-00299-t001:** Clinical characteristics of 21 patients with colonic obstruction from diverticulitis.

Mean Age (Range)	71.5 Years (50–99)
Sex (M/F)	12/9
Acute obstruction (%)	9/21 (43%)
Subacute obstruction (%)	12/21 (57%)
Severe inflammation at admission *	6/21
Moderate inflammation at admission **	9/21
Minimal inflammation at admission ***	6/21
Abdominal pain	21/21
Abdominal distension	21/21
Signs of peritoneal inflammation	6/21

* Leukocytosis; fever; signs of peritoneal in inflammation, ** Leukocytosis; fever; no signs of peritoneal inflammation, *** Minor Leukocytosis; no fever; no signs of peritoneal inflammation.

**Table 2 medicina-57-00299-t002:** Pathology of the colonic obstruction in 21 patients.

Characteristic	SEMS as Permanent Solution (4)	SEMS as Bridge to Colectomy (17)
Distance anal verge (cm) (min-max)	30 (25–35)	29 (20–38)
Length of stenosis (cm) (min-max)	6 (4–8)	6 (4–12)
Type of stent (covered/uncovered)	4/0	15/2
Length of stent (cm)	9–12	9–12
Difficult procedure	0	0
Early complications (within 7 days from post-procedure)	0	1

## Data Availability

Not applicable.

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
