# Peer review of "Self-Expandable Metal Stents for Left Sided Colon Obstruction from Diverticulitis. A Single Center Retrospective Series"

_medicina, 2021, doi:10.3390/medicina57030299_

Round 1
Reviewer 1 Report
PEER REVIEW
Dear Editor,dear Authors
I read with interest the manuscript “Self-expandable metal stents for left sided colon obstruction from diverticulitis. A useful solution in selected”
Comments:
- The main title reflects partially the major topic and content of the study, I suggest to delete “a useful solution….” Substituting it with “a retrospective single center experience”
- the abstract provide a clear delineation between the research background, objectives, materials and methods, results and conclusions
- the abstract presents the innovative and significant points related to the background, objectives, materials and methods, results and conclusions, however please define how major morbidity was defined (ASGE lexicon criteria)
Introduction:- well written, no comments
The materials and methods describe sufficiently for the results and conclusions that are presented in the preceding sections; however,
- Hinchey classification is not mentioned
- DICA classification is not mentioned, since colonoscopy was performed through a pediatric colonoscope
- Difficult procedure: on which basis?
- Distance from the anal verge: is a redundant parameter
- I consider a major flaw of the study not commenting on which basis in choosen the type of the stent (covered vs uncovered)
- ASA score of the patient is not described, it could be an useful tools about patients’ selection
Results:
- why covered stent were used as definitive treatment?
- in bridge to surgery, on which basis was choosen a covered or uncovered stent?
The discussion should be rewritten:
- I should delete all the introductory section
- I would mention the studies about the use of SEMSs in diverticular disease (i.e.. Venezia et al WJGE 2020 depict the state of the art about this topic), therefore commenting outcomes (with attention to mortality and complications)
- The conclusions drawn are not supported by the literature
- the references are appropriate, relevant, and partially up-to-date (for example, are lacking data about outcomes arising from previous studies regarding the topic)
- the tables should be corrected according to the previously mentioned suggestions, figures reflect the major findings of the study
Author Response
We respectfully submit a revision of the paper Self expandable metal stents.
The paper has been revised according to the suggestions of the reviewers. We thank their help and their taking time to review our manuscript.
Responses
Reviewer 1
Thank you for the excellent suggestions
-the title has been changed
-major morbidity was defined by the Clavien Dindo classification more often used in the surgical field. However we did not experience any III-IV-V morbidity according to Clavien Dindo classification. The level III-IV-V are considered moderate or severe complications . We experienced only minor complications. (I-II Clavien Dindo classification). (This has been specified in the text)
-we specified in the method section the reasons why covered stents were preferred.
-we specified that at CT scan there was no pericolic abscess (Hinchey classification 1,2,3)
-we added the reference by Venezia et al
-we think our study innovative. In general the literature does not support the use of stents in patients with diverticultis.
Reviewer 2
We answered the questions concerning type of stents used and the type of surgery which was generally preferred.
We have corrected the two phrases as suggested by the reviewer
Thank you again for the useful suggestions which have improved the readability of our paper.
We thank the reviewers for the time they spent to review our manuscript and to give suggestions
Reviewer 2 Report
I’ve read with great interest the paper by Antonietta Lamazza and Coworkers entitled: Self-expandable metal stents for left sided colon obstruction 3 from diverticulitis. A useful solution in selected patients.
In this work, Antonietta Lamazza and Colleagues reported a retrospective case series of SEMS placement in patient with left side colonic occlusion due to diverticoular disease.
In general the paper is well written and of clinical interest as it address a common clinical condition that can happen worldwide.
Comment:
- in the “method section” I would better clarify the type of stent used underlying that is a covered stent. Moreover I would add pictures of the stent with the technical characteristics
- what kind of surgical intervention was conducted? Laparoscopic? What about the indwelling stents? Were the stents removed before surgery or during surgery. Please, clarify.
- Please, refrase this sentence for some grammar and syntax error. Two patients are in good general conditions without symptoms at 6 months from 147 the procedure. I both patients the stent dislodged ad was expelled at 1 and 4 148 months from placement (1 patient has been already described in the previous section).
Conclusions:
Despite its retrospective design, with a small number of included patients, I think that after revision this article could be of clinical interest for the readers of the journal.
Author Response

(The authors gave the same response as above.)

Round 2
Reviewer 1 Report
Dear authors
Many thanks for improving the quality of the manuscript.
Please modify the title: A single center retrospective series
Complications should follow the ASGE lexicon, since it is an endoscopic setting
Discussion should better address the role of stenting in diverticular disease, not only quoting the studies requested
Sincerely
Author Response
Dear Editor
We respectfully submit the revised manuscript which has been corrected following the suggestions of the reviewer.
1- The title has been modified.
2-The note about the ASGE adverse events classification has been added
3- In the discussion we have added specific points about the clinical implications of SEMS placement in patients with sigmoid obstruction due to diverticultis
We thanks the reviewers for the time they took to help us to improve the quality of the paper
Respectfully yours